# A Quadratic Regression Model to Quantify Plantation Soil Factors That Affect Tea Quality



Bo Wen [1], Ruiyang Li [2], Xue Zhao [2], Shuang Ren [2], Yali Chang [3], Kexin Zhang [2], Shan Wang [2], Guiyi Guo [3,*] and Xujun Zhu [2,*]

1   College of Landscape Architecture, Nanjing Forestry University, Nanjing 210037, China; wenbo2019@njfu.edu.cn
2   College of Horticulture, Nanjing Agricultural University, Nanjing 210095, China; 14218115@njau.edu.cn (R.L.); 2018804242@njau.edu.cn (X.Z.); 2017804138@njau.edu.cn (S.R.); 2019104086@njau.edu.cn (K.Z.); 2019804235@njau.edu.cn (S.W.)
3   Henan Key Laboratory of Tea Plant Comprehensive Utilization in South Henan, Xinyang Agriculture and Forestry University, Xinyang 464000, China; changyali2019@tricaas.com
*   Correspondence: 1983240001@xyafu.edu.cn (G.G.); zhuxujun@njau.edu.cn (X.Z.)

**Abstract:** Tea components (tea polyphenols, catechins, free amino acids, and caffeine) are the key factors affecting the quality of green tea. This study aimed to relate key biochemical substances in tea to soil nutrient composition and the effectiveness of fertilization. Seventy tea samples and their corresponding plantation soil were randomly collected from Xinyang City, China. The catechins, free amino acids, and caffeine in tea were examined, as well as the soil pH, nitrate ($NO_3^-$-N), ammonium ($NH_4^+$-N), available phosphorus (AP), available potassium (AK), and soil organic matter (SOM). The ordinary kriging was employed to visualize the spatial variation characteristic by ArcGIS. A quadratic regression model was used to analyze the effects of the soil environment on the tea. The results showed that the soil pH of the study area was suitable for cultivating tea plants. The relationship between soil pH and tea polyphenols and catechins presented the U-shape curve, whereas the soil pH and $NH_4^+$-N and the free amino acids, the soil pH, and caffeine presented the inverted U-shape curve. Soil management measures could be implemented to control the soil environment for improving the tea quality. The combination of the macro metrological model with individual experimentation could help to analyze the detailed influence mechanisms of environmental factors on plant physiological processes.

**Keywords:** soil pH; soil nutrients; main chemical components in tea; spatial variation characteristic; quadratic regression model

## 1. Introduction

*Camellia sinensis* (L.) O. Ktze is one of the most important industrial crops across the world, especially in the hilly and mountain area of China [1,2]. Tea is produced by the tender leaves of *Camellia sinensis* (L.) O. Ktze. For the aroma, taste, and health advantages, tea has become a prevalent beverage in recent years [3,4]. The quality of tea is directly related to the economic benefit [5]. The tea plantation soil provides nutrients not only for the growth of tea plants but also for synthesizing functional components such as catechins, amino acids, caffeine, vitamins, and aroma-forming substances [6–10]. Thus, it is necessary to study the influence of soil nutrient concentrations on tea functional components. Some soil environmental management measures could be taken for the improvement of tea quality. Previous studies found that with sufficient N, many sugars produced by photosynthesis could be used to synthesize protein in the tea plant, resulting in an accumulation of amino acids and a decrease in polyphenols [6,11]. These results suggested that there could be a non-linear relationship between polyphenols and amino acid contents in tea and soil N supply. For the P concentration in tea plantation soil, the

deficiency of P could decrease the concentrations of total polyphenols, flavonoids, and total free amino acids [8,12]. It has been demonstrated that an adequate K could improve tolerance of drought stress and help the tea plant accumulate more catechins, amino acids, and caffeine [7]. Although sufficient N, P, and K could help tea production, excessive fertilization might damage the tea plantation soil environment and cause environmental issues such as water pollution and soil acidification [13,14]. Additionally, as the acid-base properties varied in different types of nutrients, the pH value of the plantation soil could be affected by the different components [13]. It has been demonstrated that higher or lower soil pH could inhibit growth and affect the quality and yield of tea plants [15]. Therefore, the analysis of the quantitative relationships among plantation soil pH, soil nutrients, and the main chemical component contents in tea to find the most suitable soil environment could help tea plantation management and improve the tea yield and quality.

The practical experiences of tea production show that a continuous increase of soil fertility could not limitlessly improve tea yield and quality. The hypothesis was that the key biochemical tea substances and soil nutrient composition would show a non-linear relationship. The specific objectives were to (1) examine the distribution of soil nutrients and main chemical components and (2) construct a quadratic regression model for soil pH, soil nutrients, and main chemical component concentrations in tea for analysis on the effect of soil pH and soil nutrients.

## 2. Materials and Methods

### 2.1. Study Area and Samplings

The Xinyang City of Henan Province is one of the main green tea production areas in the northern Yangtze River, China, which has a long history of tea cultivation. Xinyang Maojian Tea, which is in the top 10 most famous teas in China, originated from there. Tea plantations are widely distributed in southern Xinyang City, where the region is crisscrossed by mountains, hills, rivers, and lakes. There is a moderate climate with four distinctive seasons, as well as abundant rainfall. The average annual temperature ranges from 15.3 °C to 15.8 °C. The annual precipitation ranges from 1100 mm to 1400 mm. The soil type of the tea plantation in Xinyang City is yellow-brown soil with the parent material of weathered residual granite soil. A series of sampling sites were randomly conducted in the scope of 113°44′24″–114°16′48″ longitude and 31°51′36″–32°13′48″ latitude by the use of ArcGIS 10.3 (Figure 1). The sampling sites' longitude and latitude were recorded by GPS.

A total of 70 Xinyang local cultivars of sexual reproduction tea leave samples (one leaf and a bud) and the corresponding plantation soil (15–30 cm) were collected from random sampling sites in April 2019 (the harvest season of tea). For the soil samples, five sub-samples were collected in each sampling site. For the tea samples, each sample was comprised of the tea leaves within an approximately 100 m$^2$ range in the site.

A microwave oven (power setting: 220 V~50 Hz, 0.7 kW) was used to fix the collected fresh tea samples. In the process of fixation, the microwave oven ran twice for 1 min with a 5 min interval. The processed tea samples were put in the bake oven for 5 h for drying at 80 °C. The dried tea samples were ground into a fine powder and then passed through a 0.425 mm sieve. Polyethylene zip-lock bags were used to pack the tea samples and store them in a −20 °C refrigerator. The soil samples were air-dried at room temperature for several days and ground into a fine powder, then passed through a 0.15 mm sieve and packed in polyethylene zip-lock bags as well.

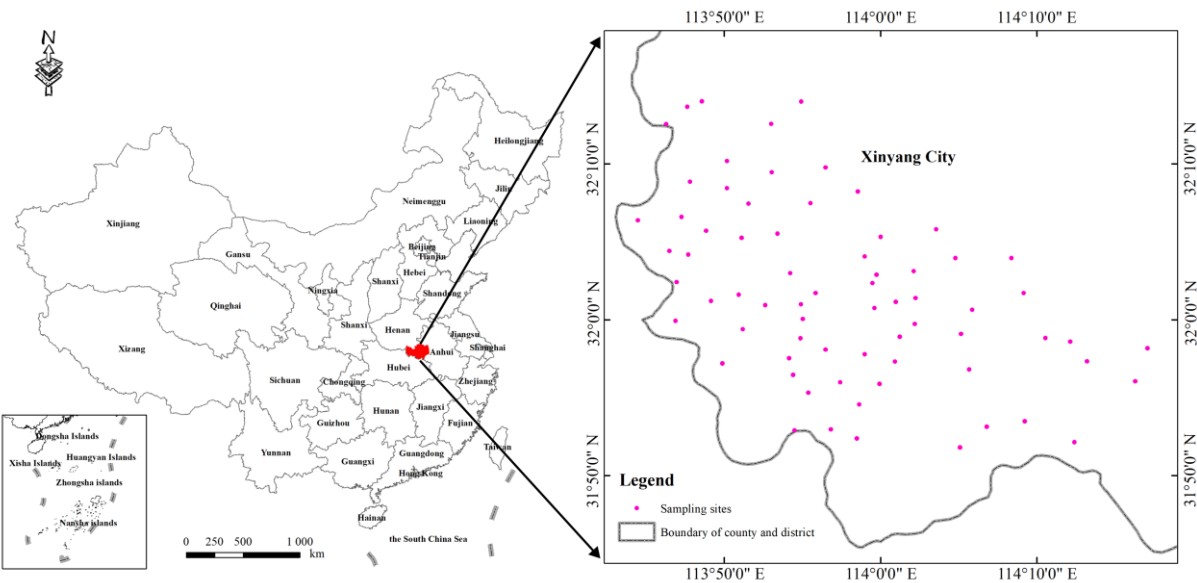

**Figure 1.** The study area and the distribution of sampling sites.

*2.2. Quantitative Determination*

Based on the methods of the International Organization for Standardization (14502-1) [16], the total tea polyphenols were determined. Briefly, 1.0 ml diluted sample extracts were transferred to 5.0 ml 10% Folin-Ciocalteu aqueous solution. A volume of 4.0 ml 7.5% sodium carbonate solution was added to the aforementioned mixed solutions. Then, the solutions were kept at room temperature for 1 h for full reaction. A spectrophotometer (Shanghai Yuanxi, Shanghai, China) was used to determine the absorbance of the solutions at 765 nm. The gallic acid standard sample was used to perform a standard curve.

According to the methods of [17], catechins and caffeine were determined by the use of Ultra Performance Liquid Chromatography (UPLC, Waters, Milford, USA). In the process of UPLC, the flow rate was set to 0.35 mL/min. The detection wavelength was set to 280 nm, and the column temperature was set to 35 °C. For the determination of free amino acids, the spectrophotometer was used according to the methods of [18]. The spectrophotometer was used to determine free amino acids at 570 nm with a 15 mm colorimetric cup, and the distilled water was used as the blank. The theanine standard sample was used to perform a standard curve.

For the soil samples, soil pH and soil $NO_3^-$-N, $NH_4^+$-N, AP, AK, and SOM concentrations were analyzed. According to the National Standard of China NY/T1377-2007, the soil pH value was measured by a pH meter (pH 197, WTW, Munich, Germany) in a water suspension using a solid-liquid ratio of 1:2.5 [19]. By the use of the UV Spectrophotometer method, the soil $NO_3^-$-N concentration was determined according to the National Standard of China (GB/T 32737-2016) [20]. For the determination of the soil $NO_3^-$-N concentration, 74.55 g potassium chloride was dissolved as an aqueous solution with 400 ml of water. Soil (40 g) was mixed in 200 ml of the potassium chloride solution. The mixture was put in a constant temperature reciprocating shaker at the frequency of $220 \pm 20$ r/min at the temperature $25 \pm 5$ °C for 1 h. Turbid liquid (60 ml) was put in the centrifugal machine conducted at a speed of 3000 r/min for 10 min. The 50 ml supernatant was taken for testing by the UV Spectrophotometer. The absorption values of the prepared extraction solution were determined at 220 nm and 275 nm. According to the methods of Xu et al. [21], $NH_4^+$-N concentrations of soil samples were determined by a continuous flow analytical system (AA3, Norderstedt, Germany). In the process of determination, 10 g of fresh soil was shaken with 100 ml of 1 mol/dm$^{-3}$ KCL for 60 min in a 250 ml jar on a reciprocating shaker at a speed of 200 rpm/min. The soil extracts were then filtered and stored at 4 °C prior to analysis using a continuous-flow analytical system within 24 h. AP concentrations of soil samples were extracted with 0.5 mol·dm$^{-3}$ NaHCO$_3$ and followed by molybdenum

blue colorimetry [22]. AK concentrations of soil samples were measured by the flame photometry method [23]. For the determination of Soil AK concentrations, 5 g of air-dried soil was shaken with 50 ml of 1 mol/dm$^{-3}$ NH$_4$OAc (pH = 7) for 30 min in a 100 ml jar on a reciprocating shaker at a speed of 200 rpm/min. The soil extracts were then filtered and analyzed using a flame photometer (FP640, INASA, China). The SOM of soil samples was determined by the potassium dichromate oxidation method [24]. All the samples were determined in three replications.

### 2.3. Geostatistical Analysis

Geostatistical analysis has been widely employed to visualize the spatial variation characteristics of plantation soil nutrients, heavy metals, agroforestry production, and meteorology [25–27]. It could be used to estimate the value of the non-sampled area by the parameters of spatial relationships. The Ordinary Kriging interpolation is the most widely applied geostatistical analysis method for the least interpolated error in different scales. In the present study, ArcGIS 10.3 was employed for the geostatistical analysis to generate a semi-variance map and describe the spatial variation characteristics of plantation soil nutrients and the main chemical components of tea.

### 2.4. Statistical Analyses

The maximum, minimum, mean, and standard deviation of soil pH and soil NO$_3^-$-N, NH$_4^+$-N, AP, AK, and SOM concentrations and the main chemical component concentrations in tea were calculated in Excel. The Quadratic Regression Model is a multiple regression model with a quadratic term, which is an effective method for the study of the effect of independent variables on dependent variables in the non-linear relationship [28]. The model has been applied in different subjects for the estimated optimal solution [29], influence factors analysis [30], and simulation [31]. In the present study, the quadratic term of independent variables could help to analyze the suitable scopes of soil pH and soil nutrients for the main chemical component accumulation in tea. Effective guidance for the plantation soil environment management could be provided for quality improvement. In the Quadratic Regression Model, an independent variable changes one unit in the case of other independent variables remaining unchanged, the dependent variable change is used to present the influence of independent variables on the dependent variable. The equation can be expressed as:

$$Y = \alpha + \sum_{i=1}^{6} \beta_i x_i + \sum_{i=1}^{6} \gamma_i x_i^2 + \varepsilon \tag{1}$$

where $Y$ stands for the concentrations of main chemical components in tea. $x_i$ (i = 1,2, . . . ,6) stands for the soil pH value and soil NO$_3^-$-N, NH$_4^+$-N, AP, AK, and SOM concentrations, and $x_i^2$ stands for the quadratic term of the pH value and soil nutrients. $\alpha$, $\beta$, and $\gamma$ represent the coefficients that need to be estimated, $\varepsilon$ is the error terms. The ordinary least squares (OLS) estimation method was applied to estimate the parameters of the regression. The multivariate statistical analyses were conducted by Stata 15.1.

## 3. Results

### 3.1. Concentrations and Distributions of Soil Nutrients and Soil pH

The soil pH and concentrations of soil nutrients of the 70 samples were summarized in Table 1. Soil pH values of the samples in Xinyang ranged from 4.22 to 6.67 with a mean of 5.23. In the analyzed soil nutrients, the soil AK concentrations were the highest, ranging from 64.14 to 405.44 mg·kg$^{-1}$, with a mean of 140.53 mg·kg$^{-1}$. The NO$_3^-$-N contents of soil samples ranged from 0.12 to 41.42 mg·kg$^{-1}$ with a mean of 11.72 mg·kg$^{-1}$, which was the lowest soil nutrient concentration of the analyzed indicators.

**Table 1.** Descriptive statistics of the soil pH and concentrations of soil nutrients.

|  | **Min** | **Max** | **Mean** | **STD** |
|---|---|---|---|---|
| pH | 4.22 | 6.67 | 5.23 | 0.56 |
| $NO_3^-$-N (mg·kg$^{-1}$) | 0.12 | 41.42 | 11.74 | 10.51 |
| $NH_4^+$-N (mg·kg$^{-1}$) | 0.52 | 117.86 | 16.65 | 23.91 |
| AP (mg·kg$^{-1}$) | 1.63 | 140.82 | 25.12 | 31.52 |
| AK (mg·kg$^{-1}$) | 64.14 | 405.44 | 140.53 | 72.67 |
| SOM (%) | 0.67 | 8.71 | 3.20 | 1.78 |

The distribution of soil pH and soil nutrient concentrations are presented in Figure 2. The soil pH values increased from the southwest to the northeast of the study area. The distributions of soil $NO_3^-$-N and $NH_4^+$-N showed a similar spatial distribution pattern, with higher concentrations in the northwest. For the spatial distribution of soil AP concentrations, most of the area had a low level, especially in the east. The soil AK showed a spatial distribution pattern with lower concentrations in the central and south than that in other locations of the area. The SOM showed higher concentrations in the west and south than in the central and northern areas. The quantity compositions and spatial distributions of tea plantations' soil nutrients differed sharply in the study area.

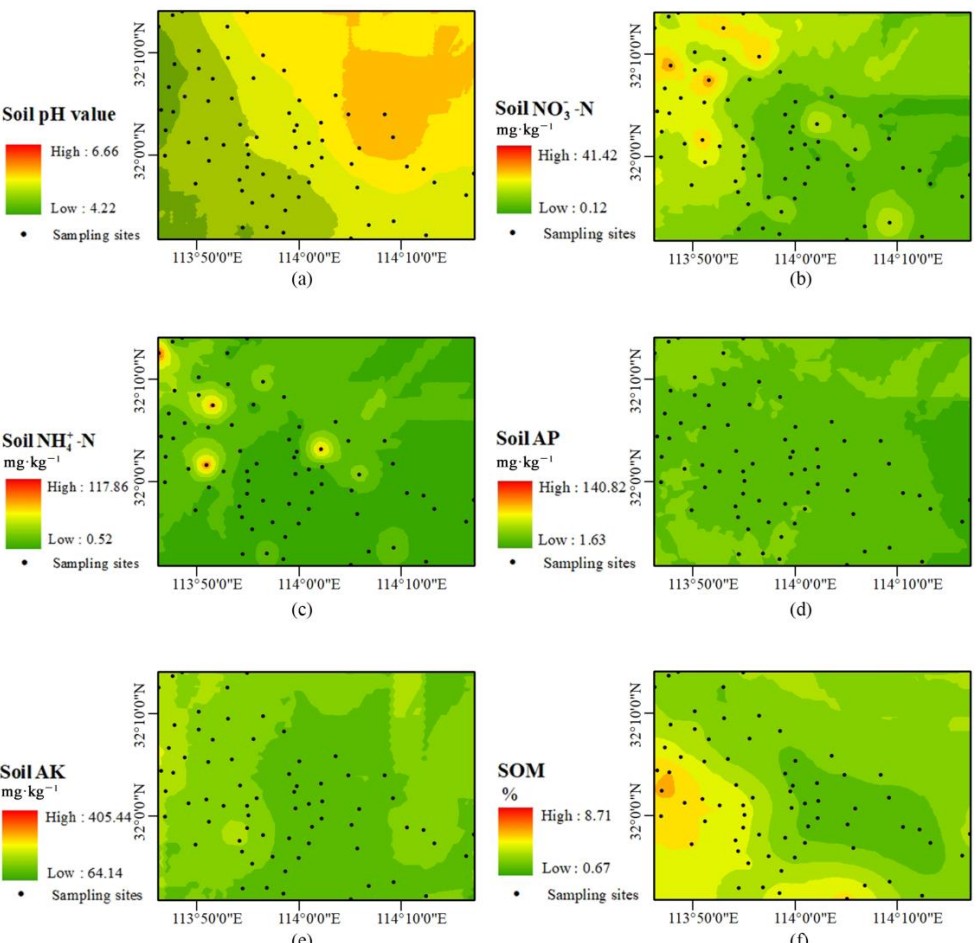

**Figure 2.** Distribution of soil pH and soil nutrients. (**a**) Distribution of soil pH. (**b**) Distribution of soil $NO_3^-$-N. (**c**) Distribution of soil $NH_4^+$-N. (**d**) Distribution of soil AP. (**e**) Distribution of soil AK. (**f**) Distribution of soil SOM.

### 3.2. Concentrations and Distributions of Main Chemical Components in Tea

The main chemical component concentrations in the 70 tea samples are summarized in Table 2. The concentrations of tea polyphenols, catechins, free amino acids, and caffeine in the 70 tea samples ranged from 161.6 mg·g$^{-1}$ to 275.1 mg·g$^{-1}$, 136.8 mg·g$^{-1}$ to 246.0 mg·g$^{-1}$, 36.0 mg·g$^{-1}$ to 55.7 mg·g$^{-1}$, and 26.3 mg·g$^{-1}$ to 46.5 mg·g$^{-1}$ with a mean value of 205.1 mg·g$^{-1}$, 183.7 mg·g$^{-1}$, 47.7 mg·g$^{-1}$, and 34.3 mg·g$^{-1}$, respectively.

**Table 2.** Descriptive statistics of the main chemical component concentrations in tea.

|  | **Min** | **Max** | **Mean** | **STD** |
|---|---|---|---|---|
| Tea polyphenols (mg·g$^{-1}$) | 161.6 | 275.1 | 205.1 | 2.78 |
| Catechins (mg·g$^{-1}$) | 136.8 | 246.0 | 183.7 | 1.22 |
| Free amino acids (mg·g$^{-1}$) | 36.0 | 55.7 | 47.7 | 0.63 |
| Caffeine (mg·g$^{-1}$) | 26.3 | 46.5 | 34.3 | 0.48 |

The distributions of the main chemical components of tea in the study area are presented in Figure 3. As the catechins are the main tea polyphenols in tea leaves, the tea polyphenols and catechins showed a similar spatial distribution pattern with higher concentrations in the east, especially in the southeast, whereas the free amino acids showed an opposite distribution pattern with a higher concentration in the northwest. The caffeine showed lower concentrations in the central region and higher concentrations in some irregular locations. From Figures 2 and 3, it seems that there are some correlations of spatial distribution characteristics between soil nutrients and main chemical components in tea.

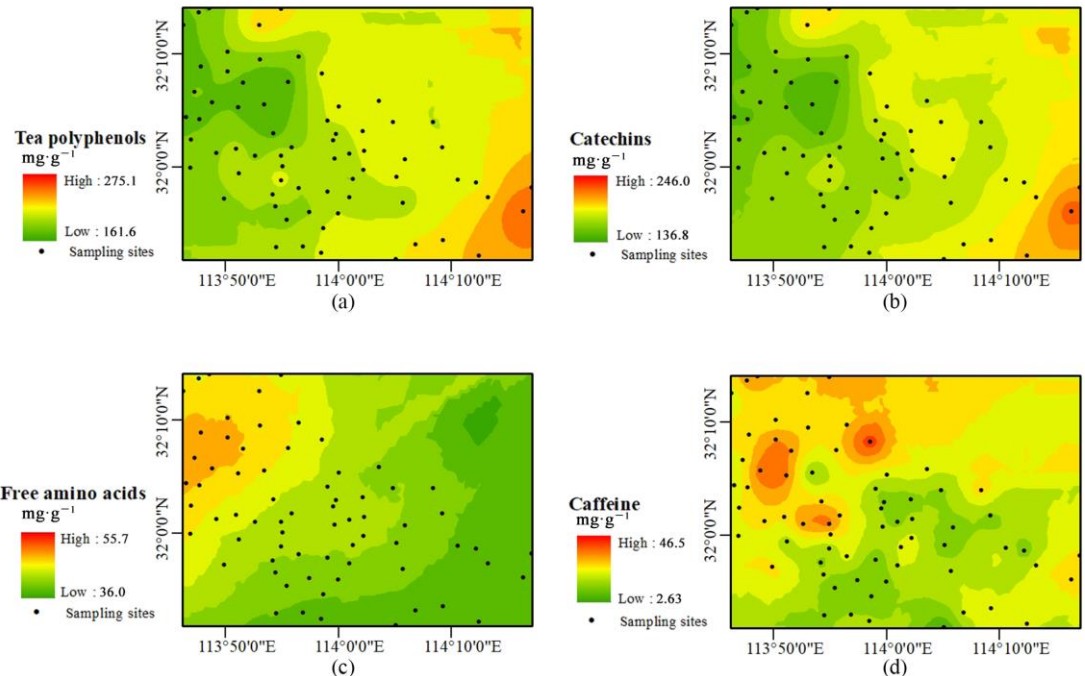

**Figure 3.** Distribution of main chemical component concentrations in tea. (**a**) Distribution of tea polyphenols. (**b**) Distribution of catechins in tea. (**c**) Distribution of free amino acids in tea. (**d**) Distribution of caffeine in tea.

### 3.3. Influence of Soil Nutrients and pH on the Main Chemical Component Concentrations in Tea

As the relationship between soil pH and nutrients and the main chemical components in tea could be non-linear, the Quadratic Regression Model was performed to assess the effects of soil nutrients and soil pH on the main chemical component concentrations in tea. The results of the influences of soil nutrients and soil pH on the main chemical components are summarized in Table 3.

**Table 3.** Analysis of the influences of soil nutrients and soil pH on the main chemical component concentrations in tea.

| | Tea Polyphenols | | Catechins | | Free Amino Acids | | Caffeine | |
|---|---|---|---|---|---|---|---|---|
| | Coef. | S.E | Coef. | S.E | Coef. | S.E | Coef. | S.E |
| pH | −30.654 * | 12.037 | −24.655 * | 10.819 | 5.489 * | 2.112 | 5.200 ** | 1.550 |
| pH (Quadratic) | 4.357 ** | 1.554 | 3.569 * | 1.406 | −0.705 ** | 0.229 | −0.570 ** | 0.204 |
| $NO_3^-$-N | −0.697 | 1.209 | −0.621 | 1.077 | 0.641 * | 0.282 | 0.288 | 0.229 |
| $NO_3^-$-N (Quadratic) | 0.018 | 0.030 | 0.014 | 0.026 | −0.011 | 0.007 | −0.004 | 0.006 |
| $NH_4^+$-N | −0.771 | 0.717 | −0.609 | 0.641 | 0.374 ** | 0.117 | 0.109 | 0.139 |
| $NH_4^+$-N (Quadratic) | 0.005 | 0.005 | 0.004 | 0.005 | −0.003 ** | 0.001 | −0.001 | 0.001 |
| AP | −0.483 | 0.467 | −0.394 | 0.416 | 0.130 | 0.084 | 0.003 | 0.055 |
| AP (Quadratic) | 0.005 | 0.004 | 0.004 | 0.004 | −0.001 * | 0.001 | −0.000 | 0.000 |
| AK | −0.180 | 0.292 | −0.163 | 0.258 | 0.075 | 0.046 | 0.014 | 0.031 |
| AK (Quadratic) | 0.001 | 0.001 | 0.000 | 0.001 | −0.001 | 0.000 | −0.000 | 0.000 |
| SOM | 0.142 | 0.942 | 0.105 | 0.830 | −0.127 | 0.170 | 0.092 | 0.119 |
| SOM (Quadratic) | −0.001 | 0.010 | −0.001 | 0.009 | 0.001 | 0.002 | −0.001 | 0.001 |
| Constant | 281.407 *** | 24.395 | 242.447 *** | 21.515 | 26.299 *** | 4.598 | 17.598 *** | 2.817 |
| N | 70 | | 70 | | 70 | | 70 | |
| F(12, 57) | 2.811 | | 2.475 | | 9.576 | | 6.876 | |
| Prob > F | 0.004 | | 0.011 | | 0.000 | | 0.000 | |
| $R^2$ | 0.211 | | 0.201 | | 0.482 | | 0.265 | |

Note: S.E. stands for robust standard errors; ***, **, and * are significant at the 0.001, 0.01, and 0.05 level.

For the models on the influence of soil nutrients and soil pH on the main chemical component concentrations in tea, all the *p*-values of the F tests were lower than 0.05, indicating that the models were valid. In the models of tea polyphenols and catechins, the results were similar, only the pH passed the significance test, and the regression coefficients of the pH were −30.654 and −24.655, whereas the quadratic regression coefficients of pH were 4.357 and 3.569, indicating that with an increase of pH, the tea polyphenols and catechin concentrations decreased first then increased. In the model of free amino acids, the pH and $NH_4^+$-N passed the significance test in one-order and quadratic terms. The results showed that the regression coefficients of the pH and $NH_4^+$-N were 5.489 and 0.374, whereas the quadratic regression coefficients were −0.705 and −0.003, respectively, indicating that with the increase of pH and $NH_4^+$-N concentration, the free amino acids concentrations increased first, then decreased. Additionally, the one-order term of $NO_3^-$-N and the quadratic term of AP passed the significance test as well; however, the regression coefficient of the AP was −0.001, demonstrating a very slight impact. The results showed that with the higher soil $NO_3^-$-N concentration, more free amino acids could be accumulated in tea. For the model of caffeine, only the pH passed the significance test, the regression coefficient was 5.200, whereas the quadratic regression coefficient was −0.570, meaning that with the increasing of pH, the caffeine concentrations increased first then decreased.

## 4. Discussion

### 4.1. Characteristics of Soil pH and Nutrients in Tea Plantations of Xinyang

The soil pH and nutrients are the foundation of tea plantation, which affects the suitability of cultivation by providing nutrients for the plant growth and the main chemical component synthesis [32,33]. In the present study, the investigated soil pH values ranged from 4.22 to 6.67, meaning that the plantation soil in Xinyang was acidified. It has been shown that soil acidification could happen naturally during tea cultivation [13,14]. Compared with the soil pH in tea plantations of other regions, such as Yunnan, Jiangsu, and Zhejiang, the soil acidification in 15–30 cm depth was slightly lower in the study area [4,34,35]. It has been found in the previous studies of cultivation experiments that when the pH is lower than 4.0 or higher than 6.5 the growth of the tea plant is inhibited and the quality of tea declines [15]. In the present study, the plantation soil pH here was suitable for tea cultivation, as the optimal soil pH for tea was 4.5–6.0, which was reported

in the previous studies. This might be one of the reasons that the study area is one of the most famous tea production areas in northern China.

Apart from the tea plantation soil pH, the soil nutrients are also very important to the yield and quality of tea. Compared with the soil nutrient contents in tea plantations of Jiangsu, Zhejiang, and Fujian, the mean $NO_3^-$-N, AP, AK, and SOM concentrations were lower, whereas the mean $NH_4^+$-N concentration was higher [6,35,36]. To promote the quality of tea, artificial measures could be taken to improve the soil fertility by supplementing $NO_3^-$-N, AP, AK, and SOM in the tea plantation soil of Xinyang.

### 4.2. Effects of Soil Nutrients and pH on Tea Quality

#### 4.2.1. For the Tea Polyphenol Contents

It has been shown that excessive fertilization inhibits tea quality and damages the soil environment [13]. In the results of the influences of soil nutrients and soil pH on tea polyphenols and catechins, only the soil pH passed the significance test. The quantitative relationship between the soil pH and tea polyphenols and catechins presented a U-shape curve (Figure 4). The quantification relationship obtained in the present study could provide some references for soil pH management. The experiments need to be further conducted for the analysis of the relationship between the soil pH and tea polyphenols and catechins in tea leaves, especially when the plantation soil pH value is out of the observed ranges in the present study. The composition of the soil nutrients are complex, and the pH properties differ among the compositions.

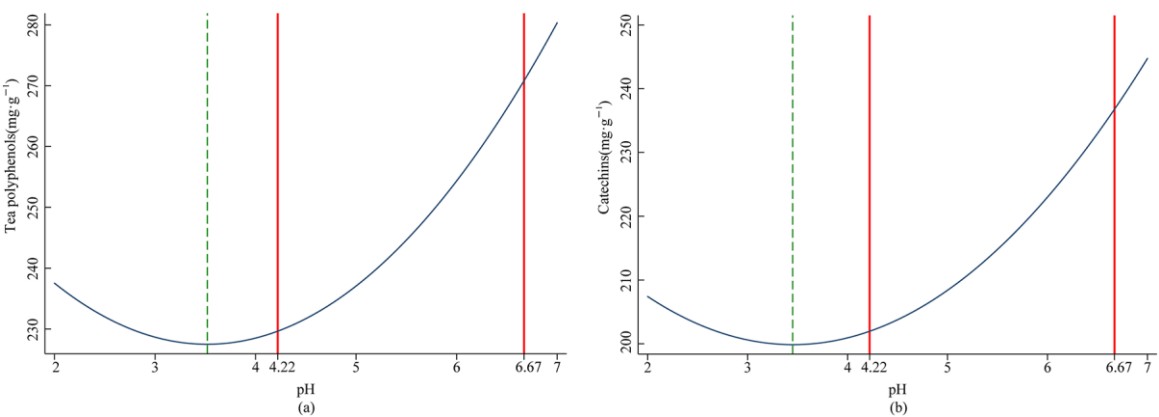

**Figure 4.** Quantitative relationship between the soil pH and tea polyphenols and catechins. (**a**) Quantitative relationship between the soil pH and tea polyphenols. (**b**) Quantitative relationship between the soil pH and catechins.

#### 4.2.2. For the Free Amino Acid Contents

The analysis of the influences of soil nutrients and soil pH on free amino acid concentrations in tea was conducted in the present study (Table 3). The concentrations of soil $NO_3^-$-N and free amino acids in tea presented a positive relationship quantitatively, which was in agreement with the previous studies [6,37]. Additionally, it was also found in the present study that the quantitative relationship between the $NH_4^+$-N and the free amino acids in tea presented an inverted U-shaped curve (Figure 5). As the quadratic regression coefficient of $NH_4^+$-N was −0.003, the curve reached its apex and began to drop but the decline was less. Taking the soil $NO_3^-$-N and $NH_4^+$-N together, the increase of the effective N supplied could increase the accumulation of free amino acids in tea within a certain limit. The results of the data analyses were in line with the previous studies, which were conducted in physiological and biochemical experiments [6,11,37].

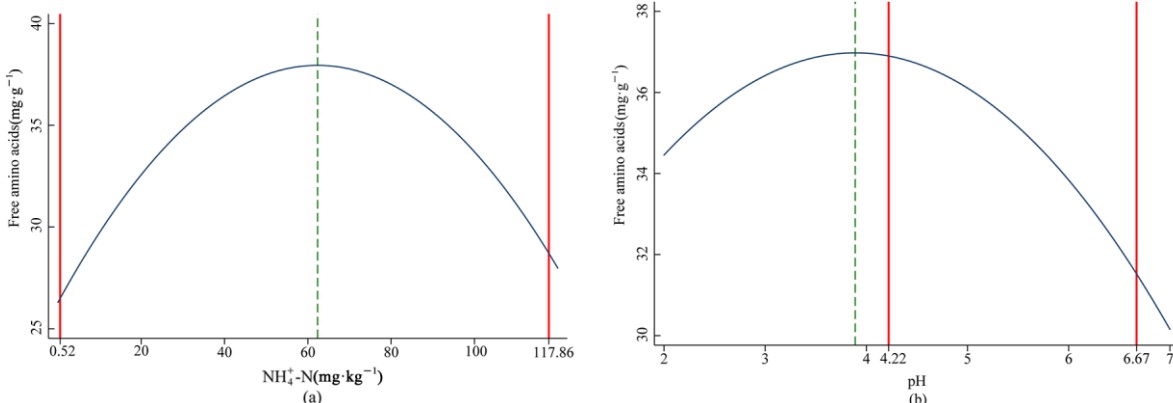

**Figure 5.** Quantitative relationship between the free amino acids and $NH_4^+$-N and soil pH. (**a**) Quantitative relationship between the free amino acids and $NH_4^+$-N. (**b**) Quantitative relationship between free amino acids and soil pH.

For the effect of soil pH on free amino acids in tea, the quantitative relationship between the soil pH and the free amino acids in tea presented an inverted U-shaped curve as well (Figure 5), which was opposite to the effect of soil pH on polyphenols in tea. The results could be associated with the process of regulation in synthesizing secondary metabolism in tea plants. In the lower pH values, the free amino acid concentrations increased with the pH values increasing, whereas the tea polyphenols concentrations declined with the pH values increasing. However, the characteristic of the free amino acids and tea polyphenols concentrations was opposite when the pH values were in the higher levels. It has also been found in previous studies that there was a dynamic relationship between the accumulations of tea polyphenols and free amino acids, especially theanine in the tea plant [2,38]. In the condition of sufficient N supply, the carbon-containing compounds, which were produced by photosynthesis, were used to synthesize protein, and the process of sugars to polyphenols was restricted [39]. Combination of the results in the present study and the findings of previous studies, measures such as controlling the soil effective N supply, managing the SOM of tea plantation, and adjusting the soil pH within an appropriate limit directly could be considered to obtain higher free amino acids in tea.

### 4.2.3. For the Caffeine Contents

The analysis of the influences of soil pH on caffeine concentrations in tea was also conducted in the present study (Table 2). The result showed that the quantitative relationship between the soil pH and the caffeine in tea was an inverted U-shaped curve (Figure 6). This was different from the previous studies, which have reported that the tea from the different cultivation locations, altitudes, slope of tea plantations, and intercropping patterns had no significant difference in caffeine contents [2,5,36]. The caffeine concentrations in tea leaves were more likely related to the cultivars, leaves age of raw materials, and climatic factors during plant growing [38,40]. As was found in the present study, the caffeine contents in tea were influenced by soil pH, and the soil nutrients had no significant effects in the metrological model. The soil pH has an intense influence on tea plant synthesis of secondary metabolites; controlling the pH of soil could also help to obtain suitable caffeine contents for the good flavor and taste.

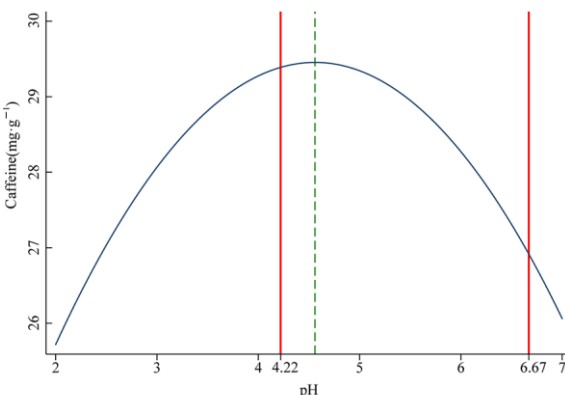

**Figure 6.** Quantitative relationship between the caffeine and soil pH.

## 5. Conclusions

The investigated tea plantation soil samples' pH values ranged from 4.22 to 6.67 and the result of Ordinary Kriging of the soil pH value showed an acid pattern, which was suitable for the tea planting in this area. Although the tea polyphenol and catechin concentrations increased with the pH values in the observed ranges, the Quadratic Regression Models showed the tendency that with the increase of soil pH values, the tea polyphenols and catechins concentrations reduced first, but increased later. Theoretically, their levels were lowest when the soil pH values ranged from 3 to 4. In the models for the free amino acids, the levels got the highest when the $NH_4^+$-N concentrations were around 60 mg·kg$^{-1}$ and the soil pH values around 4, respectively. For the model of caffeine, the levels were highest when the pH values were around 4.5. As all the analyses were carried out based on the 70 observed samples, the experiments need to be further conducted when the soil nutrients and soil pH on the main chemical component concentrations are out of the observed ranges in the present study. Accordingly, these findings could guide tea plantation soil management for the improvement of tea quality. The combination of the macro metrological model with the individual physiological and biochemical experiments could help to analyze the detailed influence mechanisms of environmental factors on plant physiological processes.

**Author Contributions:** Conceptualization, X.Z. (Xujun Zhu) and G.G.; methodology, B.W.; software, R.L.; formal analysis, R.L., X.Z. (Xue Zhao), S.R; investigation, S.R., Y.C., K.Z.; resources, G.G.; data curation, Y.C.; writing—original draft preparation, B.W.; writing—review and editing, X.Z. (Xujun Zhu); visualization, S.W.; supervision, G.G.; project administration, X.Z. (Xujun Zhu); funding acquisition, X.Z., G.G. All authors have read and agreed to the published version of the manuscript.

**Funding:** This research was funded by The National Natural Science Foundation of China (grant number 31870680) and the Open Fund of Henan Key Laboratory of Tea Plant Comprehensive utilization in South Henan (grant number HNKLTOF2018002).

**Institutional Review Board Statement:** Not applicable.

**Informed Consent Statement:** Not applicable.

**Data Availability Statement:** Not applicable.

**Conflicts of Interest:** The authors declare no conflict of interest.

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
