# Peer review of "A Quadratic Regression Model to Quantify Plantation Soil Factors That Affect Tea Quality"

_agriculture, doi:10.3390/agriculture11121225_

Round 1

Reviewer 1 Report

The work describes in an interesting and clear manner selected properties of soils from the tea growing region in Henan Province, the content of polyphenols, catechins, free amino acids and caffeine in tea leaves, and using regression equations - of the relationship between them and soil properties. Given the economic importance of tea and the amount of global consumption, identifying factors that influence the quality of the stimulant would be of great help in the production process. The work takes up this challenge, however, some conclusions require confirmation in additional research (details in the comments in the text).
The text is carefully prepared, the work is well documented. Slight additions to the methodological part are indicated. The results are presented clearly and legibly. It is recommended to check the presented calculation values ​​in table 3. Detailed remarks are included in the form of comments in the text of the paper. 

Author Response

Thank you very much for the review and for giving us an opportunity to revise our manuscript. The comments and suggestions are very important to us for improving our manuscript.

Reviewer 2 Report

The authors must improve substantially the materials and methods section. The authors must provide:

  1. The geology (parent materials) of the studey area
  2. The soil types (classification according to Soil Taxonomy or WRB)
  3. The climate parameters
  4. Describe how many leaves per cultivar was collected

Moreover, what is the effect of the different tea cultivars in secondary metabolites in relation to the results of the study?

Author Response

Thank you very much for giving us an opportunity to revise our manuscript. We checked the grammatical, organizational, and type-writing again carefully one sentence by one sentence. The logical relationship between sentences is improved by reorganizing the language of our manuscript. The contents of the modifications are listed in the attachment.

Reviewer 3 Report

This is a really nice paper reporting a quadratic regression model that can be used to quantify plantation soil factors that affect tea quality.

I'd suggest to change the title in: "A quadratic regression model to quantify plantation soil factors that affect tea quality"

As for the paper, I really enjoyed reading it. I'd suggest to better specify the sampling procedures. 

When the Authors stated: "A total of 70 Xinyang local cultivar's tea leaves samples (one leaf and a bud) and the corresponding plantation soil (15-30 cm) were collected from the random sampling sites in April 2019 (the harvest season of tea)." What are these cultivars? The contents of phytochemical can vary across cultivars because of the genotype. Also, what are the soil characteristics? Usually soils are divided in 12 orders: Entisols, Inceptisols, Andisols, Mollisols, Alfisols, Spodosols, Ultisols, Oxisols, Gelisols, Histosols, Aridisols, and Vertisols.

Knowing the cultivars and the soil orders will help in creating more accurate regression and correlation curves.

Author Response

Thank you very much for the review and for giving us an opportunity to revise our manuscript. We checked the grammatical, organizational, and type-writing again carefully one sentence by one sentence. The logical relationship between sentences is improved by reorganizing the language of our manuscript. The contents of the modifications are listed in the attachment.

Round 2

Reviewer 2 Report

The manuscript has been revised according to reviewer comments.

Reviewer 3 Report

The manuscript has been substantially improved.

This manuscript is a resubmission of an earlier submission. The following is a list of the peer review reports and author responses from that submission.

Round 1

Reviewer 1 Report

This an exploratory analysis of the relationship between soil nutrients and tea quality indices. Provide hypotheses. Statistical analyses are inherently biased. Separate the Resuts and Discussion section. In Discusison, you can focus on the most important points.

Introduction

l. 14-16: you should start with two sentences that first shows the importance of tea components in the region under study and secondly address problem to solve.

l. 17: what do you mean by “quantitative relationship”? I think that just saying “this study aimed to relate tea key biochemical substances to soil nutrient composition and effectiveness of fertilization” would suffice

l. 21-22: it seems that “quadratic” refers to the kriging procedure rather than crop response. Later, I realized that it refers to regression analysis.

l. 29: metrological?

l. 43: references 6 and 7 report on N and K only, not all nutrients

l. 44: what do you mean by “structures”? Do you mean nutrient balances, soil physical structure?

l. 44-45: only soil is discussed in preceding sentences. What is soil environmental mamagement?

l. 46-48: rewrite correctly

l. 66: why didn’t you analyze foliar nutrients where key tea components are synthesized?

l. 67-68: strange sentence

l. 71: the quadratic tends to recommend over-fertilizing agricultural crops. The references provided are little useful.

l. 80-85: move to M&M or elsewhere

l. 87-89: What is the main objective of the study? The specific objectives were to (1) clarify the distribution of soil nutrients and main chemical components in tea in Xinyang and (2) construct the quadratic regression model for soil pH, soil nutrients, and main chemical components concentration in tea; and (3) analyze the effect of soil pH and soil nutrients on main chemical components concentration of tea. Specific objectives (sic) (2) and (3) appears to be the same. You should rather present the hypotheses to be tested, so you can formuate a quantative conclusion based on the results of statistical analyses so that a conclusion can be reached based on the results of statistical analysis.

M&M

Figure 1 is little informative. Locate Xinyang city in China first, then focus to the sampling procedure.

l. 98: were there 70 cultivars or 70 samples of some cultivars (mention number and names of cultivars)?

l. 99: sampling soils at depth of 15-30 cm is uncommon. Justify this sampling procedure. In soil fertility studies, we usually analyze the 0-15 or 0-20 cm soil layers.

l. 100: five subsamples is small. In general, we composite 20 foliar samples per plot.

l. 101-103: what is the purpose of microwaving foliar samples? How many 5-min intervals between 1-min microwavings? Oven drying at 80°C is too high. Volatile aromas will be lost. How long was oven-drying? If you freeze at -20°C, it is preferable to freeze-dry the leaves. Why didn’t you do that?

l. 106: mention air temperature for soil drying

l. 118: spell out UPLC in length when first mentioning

l. 126: Jin et al. (2019) and Xu et al. (2020) are not about China standards. 

l. 133: describe briefly the extraction methods for NO3-N, NH4-N, AP and AK. Results depend on methods that may vary widely among countries. It seems that AP was Olsen-P. What about AK, NO3-N and NH4-N?

l. 149-150: correlation… correlation

l. 152-157: was the relationship between the dependent and independent variables established one at the time? This assumes heroically that all features but the ones being varied are equal or at optimum levels. Why not use canonical analysis?

Results

l. 166-168: SOM is not a nutrient. Such order is agronomically meaningless because each one impacts crop quality differently.

l. 200: you must conduct a full quantitative comparison.

Table 1: CV of pH values is meaningless because pH has a log basis. If you want to compare the variations in soil features, log-transform them first then compute standard deviations.

l. 232: this does not indicate that the model is valid but that such relationship is significant. Indeed, compositional data such as the results of chemical and biochemical analyses are intrinsically multivariate and impact on each other within the closed space of the entity (here tea leaf). The correlations and regressions are biased by interactions between tea components, hence tests of significance are wrong (Table 3). Bias can be tackled by using log ratio transformations. Moreover, the multivariate character of tea components as part of the leaf entity can be described by tools of multivariate analysis such as PCA and canonical analysis using log ratios as variables.

l. 255, 280, 297: show U curves.

Table 4 is not very relevant to the study nor interpretable in terms of management because many factors (e.g., l. 201, 289-290, 299-302) impact the results of soil tests.

l. 282-285: this a short hop

Conclusion

Too general and little informative.

Author Response

Thank you very much for giving us an opportunity to revise our manuscript. We tried our best to revise our manuscript. In the process of the revision, we also learned a lot. The grammatical, organizational, and type-writing has been check again carefully one sentence by one sentence. The logical relationship between sentences is improved by reorganizing the language of our manuscript. 

Reviewer 2 Report

Dear Authors,

The subject of the study is interesting and topical, with high scientific and practical importance.

The introduction is presented correctly, in accordance with the subject. Numerous scientific articles, in concordance to the topic of the study, were consulted.

Methodology of the study was clearly presented, and appropriate to the proposed objectives.

The obtained results are important and have been analyzed and interpreted correctly, in accordance with the current methodology.

The discussions are appropriate, in the context of the results, and was conducted compared to other studies in the field.

The scientific literature, to which the reporting was made, is recent and representative in the field.

Some suggestions and corrections were made in the article.

The following aspects are brought to the attention of the authors.

1.

Title of the article

A title suggestion

“Quadratic regression model to quantify the plantation soil factors that affecting tea quality”

Instead of

“Quadratic regression model for the quantification evaluation of plantation soil factors affecting tea quality”

It is up to the authors to decide.

2.

It is recommended to use the Equation Editor to write the ionic forms correctly

“NO3⁻-N” instead of current form

“NH4-N” instead of current form

It is “NO-“ and not “3-

respective,

“NH+” and not “4+

For this reason, it is more appropriate to use the Equation Editor to write the ionic forms correctly.

3.

“113° 74’ - 114° 28’ Longitude and 31° 86’ - 32° 23’ Latitude” instead of “113.74°-114.28° longitude and 31.86°-32.23° latitude”

In this way, it is a notation much more in concordance with the map in Figure 1.

4.

In the correlation table it is not recommended to write the values that express the autocorrelation

eg page 8, Table 8,

pH - pH, r = 1.000; similar for the other cases.

It was suggested to eliminate those values.

5.

Discussions chapter

The Discussions chapter is missing.

According to the Instructions for Authors, and Microsoft Word template, Agriculture journal, Results is a separate chapter, and Discussions is also a separate chapter.

There is a provision regarding the Discussion chapter, such as: "This section may be combined with Results."

In this article, the results obtained are presented combined with discussions.

It is the decision of the authors on the final form of presentation.

6.

The names of the species are recommended to be written in Font style Italic.

eg

Page 1, rows 36, 38

“(Camellia sinensis)” instead of “(Camellia sinensis)”

Page 10, row 346

“(Camellia sinensis)” instead of “(Camellia sinensis)”

Page 11, row 382

Erythrina verna” instead of ”Erythrina verna”

Page 11, row 399

Robinia pseudoacacia” instead of ”Robinia pseudoacacia”

Page 12, row 411

“(Camellia sinensis L.)” instead of “(Camellia sinensis L.)”

Page 12, rows 425, 428

“(Camellia sinensis)” instead of “(Camellia sinensis)”

7.

References chapter

a)

Abbreviated Journal Name

Eg.

Page 10

"J. Sci. Food Agric." instead of “Journal of the science of food and agriculture”

"J. Plant Nutr." instead of “Journal of Plant Nutrition”

b)

According to the Instructions for Authors, and Microsoft Word template, Agriculture journal, it is not recommended that every word in the title of an article be capitalized.

eg

Page 11, rows 372, 373

“Spatial variability analysis of within-field winter wheat nitrogen and grain quality using canopy fluorescence sensor measurements”

instead of

”Spatial Variability Analysis of Within-Field Winter Wheat Nitrogen and Grain Quality Using Canopy Fluorescence Sensor Measurements”

Similarly,

Page 12, rows 435, 436

Author Response

Thank you very much for giving us an opportunity to revise our manuscript. We checked the grammatical, organizational, and type-writing again carefully one sentence by one sentence. The logical relationship between sentences is improved by reorganizing the language of our manuscript. 

Round 2

Reviewer 1 Report

Authors made some useful corrections but did not fully address my concerns.

l. 41: delete 'According to the results'. Such phrase often occurs throughout the Ms but the reader suspects that the arguments are based on results.

Q3: l. 63-74 should be moved to the 'Statistical analysis section'. L. 59-62 suffice to introduce the relationship searched for.

Q8: isn't it trivial?

Q10: l. 63-74 should be moved to the 'Statistical analysis section'. L. 59-62 suffice to introduce the relationship searched for.

l. 76-78: poorly written. 'Xinyang' has not been introduced earlier. Alternatiely, you can just delete it because mentioned in the M&M section below.

l. 87: there

Fig. 1: the study area seems to be in Central rather than Northern China.

l. 93: mention the name of the cultivar

l. 95: justify the soil sampling in layer 15-30 cm (see response to Q15)

l. 99: rephrase

l. 151-154: Eq. 1 refers to one nutrient at the time or several nutrients combined, each one showing quadratic behavior? It is not clear whether the regression was multivariate (several soil properties) as stated l. 153 or univariate (one property at the time) as in Eq. 1. Clarify this.

Table 1: even if CV is used in previous papers, it does not mean that it was correct. The log have special properties that make CV meaningless. Just mention STD in Table 1.

Figs. 2-3: use more contrasting colors. It looks strange that interpolations occur where there were not a single sampling point.

Q27: this was to indicate that applying linear statistics to intrinsically related data adding up to a bounded sum (here leaf dry matter) producing spurious correlations is biased and can just be interpreted with care as exploratory analysis. See J. Aitchison, “The Statistical Analysis of Compositional Data,” Chapman and Hall, London, 1986.

Q28: curves in Figs 4-5-6 are little informative is the distribution of the points and the quadratic equations and their R2 values are not presented. Ranges of soil properties should be provided. Otherwise, those figures are useless. In addition, no model is provided for N, P and C appeared to impact tea composition significantly (Table 3). In Table 3, it is  not clear whether F and R2 refer to a multivariate regression with quadratic components or not.

Q31: the conclusion must be quantitative because it was the objective pursued by the authors. What is the novelty of concluding on general statements? Indeed, the conclusion looks like an introduction. 

Author Response

Dear Reviewer,

Thank you very much for your kind suggestions. All authors appreciated your efficiency and excellent work. After study your comments carefully, the manuscript has been revised. The contents of the modifications are listed in the attachment.
